# Small-Variance Asymptotics for Hidden Markov Models

**Anirban Roychowdhury, Ke Jiang, Brian Kulis**
Department of Computer Science and Engineering
The Ohio State University
roychowdhury.7@osu.edu, {jiangk,kulis}@cse.ohio-state.edu

## Abstract

Small-variance asymptotics provide an emerging technique for obtaining scalable combinatorial algorithms from rich probabilistic models. We present a small-variance asymptotic analysis of the Hidden Markov Model and its infinite-state Bayesian nonparametric extension. Starting with the standard HMM, we first derive a "hard" inference algorithm analogous to $k$-means that arises when particular variances in the model tend to zero. This analysis is then extended to the Bayesian nonparametric case, yielding a simple, scalable, and flexible algorithm for discrete-state sequence data with a non-fixed number of states. We also derive the corresponding combinatorial objective functions arising from our analysis, which involve a $k$-means-like term along with penalties based on state transitions and the number of states. A key property of such algorithms is that—particularly in the nonparametric setting—standard probabilistic inference algorithms lack scalability and are heavily dependent on good initialization. A number of results on synthetic and real data sets demonstrate the advantages of the proposed framework.

## 1  Introduction

Inference in large-scale probabilistic models remains a challenge, particularly for modern "big data" problems. While graphical models are undisputedly important as a way to build rich probability distributions, existing sampling-based and variational inference techniques still leave some applications out of reach.

A recent thread of research has considered *small-variance asymptotics* of latent-variable models as a way to capture the benefits of rich graphical models while also providing a framework for designing more scalable combinatorial optimization algorithms. Such models are often motivated by the well-known connection between mixtures of Gaussians and $k$-means: as the variances of the Gaussians tend to zero, the mixture of Gaussians model approaches $k$-means, both in terms of objectives and algorithms. This approach has recently been extended beyond the standard Gaussian mixture in various ways—to Dirichlet process mixtures and hierarchical Dirichlet processes [8], to non-Gaussian observations in the nonparametric setting [7], and to feature learning via the Beta process [5]. The small-variance analysis for each of these models yields simple algorithms that feature many of the benefits of the probabilistic models but with increased scalability. In essence, small-variance asymptotics provides a link connecting some probabilistic graphical models with non-probabilistic combinatorial counterparts.

Thus far, small-variance asymptotics has been applied only to fairly simple latent-variable models. In particular, to our knowledge there has been no such analysis for sequential data models such as the Hidden Markov Model (HMM) nor its nonparametric counterpart, the infinite-state HMM (iHMM). HMMs are one of the most widely used probabilistic models for discrete sequence data, with diverse applications including DNA sequence analysis, natural language processing and speech recognition [4]. The HMMs consist of a discrete hidden state sequence that evolves according

to Markov assumptions, along with independent observations at each time step depending on the hidden state. The learning problem is to estimate the model given only the observation data.

To develop scalable algorithms for sequential data, we begin by applying small-variance analysis to the standard parametric HMM. In the small-variance limit, we obtain a penalized k-means problem where the penalties capture the state transitions. Further, a special case of the resulting model yields segmental k-means [9]. For the nonparametric model we obtain an objective that effectively combines the asymptotics from the parametric HMM with the asymptotics for the Hierarchical Dirichlet Process [8]. We obtain a $k$-means-like objective with three penalties: one for state transitions, one for the number of reachable states out of each state, and one for the number of total states. The key aspect of our resulting formulation is that, unlike the standard sampler for the infinite-state HMM, dynamic programming can be used. In particular, we describe a simple algorithm that monotonically decreases the underlying objective function. Finally, we present results comparing our non-probabilistic algorithms to their probabilistic counterparts, on a number of real and synthetic data sets.

**Related Work.** In the parametric setting (i.e., the standard HMM), several algorithms exist for maximum likelihood (ML) estimation, such as the Baum-Welch algorithm (a special instance of the EM algorithm) and the segmental $k$-means algorithm [9]. Infinite-state HMMs [3, 12] are nonparametric Bayesian extensions of the finite HMMs where hierarchical Dirichlet process (HDP) priors are used to allow for an unspecified number of states. Exact inference in this model is intractable, so one typically resorts to sampling methods. The standard Gibbs sampling methods [12] are notoriously slow to converge and cannot exploit the forward-backward structure of the HMMs. [6] presents a Beam sampling method which bypasses this obstacle via slice sampling, where only a finite number of hidden states are considered in each iteration. However, this approach is still computationally intensive since it works in the non-collapsed space. Thus infinite-state HMMs, while desirable from a modeling perspective, have been limited by their inability to scale to large data sets—this is precisely the situation in which small-variance asymptotics has the potential to be beneficial.

Connections between the mixture of Gaussians model and k-means are widely known. Beyond the references discussed earlier, we note that a similar connection relating probabilistic PCA and standard PCA was discussed in [13, 10], as well as connections between support vector machines and a restricted Bayes optimal classifier in [14].

## 2 Asymptotics of the finite-state HMM

We begin, as a warm-up, with the simpler parametric (finite-state) HMM model, and show that small-variance asymptotics on the joint log likelihood yields a penalized k-means objective, and on the EM algorithm yields a generalized segmental k-means algorithm. The tools developed in this section will then be used for the more involved nonparametric model.

### 2.1 The Model

The Hidden Markov Model assumes a hidden state sequence $\mathcal{Z} = \{z_1, \ldots, z_N\}$ drawn from a finite discrete state space $\{1, \ldots, K\}$, coupled with the observation sequence $\mathcal{X} = \{\boldsymbol{x}_1, \ldots, \boldsymbol{x}_N\}$. The resulting generative model defines a probability distribution over the hidden state sequence $\mathcal{Z}$ and the observation sequence $\mathcal{X}$. Let $T \in \mathbb{R}^{K \times K}$ be the stationary transition probability matrix of the hidden state sequence with $T_{i\cdot} = \boldsymbol{\pi}_i \in \mathbb{R}^K$ being a distribution over the latent states. For clarity of presentation, we will use a binary 1-of-K representation for the latent state assignments. That is, we will write the event of the latent state at time step $t$ being $k$ as $z_{tk} = 1$ and $z_{tl} = 0 \quad \forall l = 1 \ldots K, l \neq k$. Then the transition probabilities can be written as $T_{ij} = \Pr(z_{tj} = 1 | z_{t-1,i} = 1)$. The initial state distribution is $\boldsymbol{\pi}_0 \in \mathbb{R}^K$. The Markov structure dictates that $z_t \sim \text{Mult}(\boldsymbol{\pi}_{z_{t-1}})$, and the observations follow $\boldsymbol{x}_t \sim \Phi(\theta_{z_t})$. The observation density $\Phi$ is assumed invariant, and the Markov structure induces conditional independence of the observations given the latent states.

In the following, we present the asymptotic treament for the finite HMM with Gaussian emission densities $\Pr(\boldsymbol{x}_t | z_{tk} = 1) = \mathcal{N}(\boldsymbol{x}_t | \boldsymbol{\mu}_k, \sigma^2 I_d)$. Here $\theta_{z_t} = \boldsymbol{\mu}_k$, since the parameter space $\Theta$ contains only the emission means. Generalization to exponential family emission densities is straightforward[7]. At a high level, the connection we seek to establish can be proven in two ways. The first approach is to examine small-variance asymptotics directly on the the joint probability distribution of the HMM, as done in [5] for clustering and feature learning problems. We will primarily

focus on this approach, since our ideas can be more clearly expressed by this technique, and it is independent of any inference algorithm. The other approach is to analyze the behavior of the EM algorithm as the variance goes to zero. We will briefly discuss this approach as well, but for further details the interested reader can consult the supplementary material.

### 2.1.1 Exponential Family Transitions

Our main analysis relies on appropriate manipulation of the transition probabilities, where we will use the bijection between exponential families and Bregman divergences established in [2]. Since the conditional distribution of the latent state at any time step is multinomial in the transition probabilities from the previous state, we use the aforementioned bijection to refactor the transition probabilities in the joint distribution of the HMM into a form than utilizes Bregman divergences. This, with an appropriate scaling to enable small-variance asymptotics as mentioned in [7], allows us to combine the emission and transition distributions into a simple objective function.

We denote $T_{jk} = \Pr(z_{tk} = 1 | z_{t-1,j} = 1)$ as before, and the multinomial distribution for the latent state at time step t as

$$\Pr(\boldsymbol{z}_t | z_{t-1,j} = 1) = \prod_{k=1}^{K} T_{jk}^{z_{tk}}. \tag{1}$$

In order to apply small-variance asymptotics, we must allow the variance in the transition probabilities to go to zero in a reasonable way. Following the treatment in [2], we can rewrite this distribution in a suitable exponential family notation, which we then express in the following equivalent form:

$$\Pr(\boldsymbol{z}_t | z_{t-1,j} = 1) = \exp(-d_\phi(\boldsymbol{z}_t, \boldsymbol{m}_j)) b_\phi(\boldsymbol{z}_t), \tag{2}$$

where the Bregman divergence $d_\phi(\boldsymbol{z}_t, \boldsymbol{m}_j) = \sum_{k=1}^{K} z_{tk} \log\left(z_{tk}/T_{jk}\right) = \mathrm{KL}(\boldsymbol{z}_t, \boldsymbol{m}_j)$, $\boldsymbol{m}_j = \{T_{jk}\}_{k=1}^{K}$ and $b_\phi(\boldsymbol{z}_t) = 1$. See the supplementary notes for derivation details. The prime motivation for using this form is that we can appropriately scale the variance of the exponential family distribution following Lemma 3.1 of [7]. In particular, if we introduce a new parameter $\hat{\beta}$, and generalize the transition probabilities to be

$$\Pr(\boldsymbol{z}_t | z_{t-1,j} = 1) = \exp(-\hat{\beta} d_\phi(\boldsymbol{z}_t, \boldsymbol{m}_j)) b_{\tilde{\phi}}(\boldsymbol{z}_t),$$

where $\tilde{\phi} = \hat{\beta}\phi$, then the mean of the distribution is the same in the scaled distribution (namely, $\boldsymbol{m}_j$) while the variance is scaled. As $\hat{\beta} \to \infty$, the variance goes to zero.

The next step is to link the emission and transition probabilities so that the variance is scaled appropriately in both. In particular, we will define $\beta = 1/2\sigma^2$ and then let $\hat{\beta} = \lambda\beta$ for some $\lambda$. The Gaussian emission densities can now be written as $\Pr(\boldsymbol{x}_t | z_{tk} = 1) = \exp(-\beta\|\boldsymbol{x}_t - \boldsymbol{\mu}_k\|_2^2) f(\beta)$ and the transition probabilities as $\Pr(\boldsymbol{z}_t | z_{t-1,j} = 1) = \exp(-\lambda\beta d_\phi(\boldsymbol{z}_t, \boldsymbol{m}_j)) b_{\tilde{\phi}}(\boldsymbol{z}_t)$. See [7] for further details about the scaling operation.

### 2.1.2 Joint Probability Asymptotics

We now have all the background development required to perform small-variance asymptotics on the HMM joint probability, and derive the segmental k-means algorithm. Our parameters of interest are the $Z = [\boldsymbol{z}_1, ..., \boldsymbol{z}_N]$ vectors, the $\boldsymbol{\mu} = [\boldsymbol{\mu}_1, ..., \boldsymbol{\mu}_K]$ means, and the transition parameter matrix $T$. We can write down the joint likelihood by taking a product of all the probabilities in the model:

$$p(\mathcal{X}, \mathcal{Z}) = p(\boldsymbol{z}_1) \prod_{t=2}^{N} p(\boldsymbol{z}_t | \boldsymbol{z}_{t-1}) \prod_{t=1}^{N} \mathcal{N}(\boldsymbol{x}_t | \boldsymbol{\mu}_{z_t}, \sigma^2 I_d),$$

With some abuse of notation, let $\boldsymbol{m}_{\boldsymbol{z}_{t-1}}$ denote the mean transition vector given by the assignment $\boldsymbol{z}_{t-1}$ (that is, if $z_{t-1,j} = 1$ then $\boldsymbol{m}_{\boldsymbol{z}_{t-1}} = \boldsymbol{m}_j$). The exp-family probabilities above allow us to rewrite this joint likelihood as

$$p(\mathcal{X}, \mathcal{Z}) \propto \exp\left[-\beta\left(\sum_{t=1}^{N} \|\boldsymbol{x}_t - \boldsymbol{\mu}_{z_t}\|_2^2 + \lambda \sum_{t=2}^{N} \mathrm{KL}(\boldsymbol{z}_t, \boldsymbol{m}_{\boldsymbol{z}_{t-1}})\right) + \log p(\boldsymbol{z}_1)\right]. \tag{3}$$

To obtain the corresponding non-probabilistic objective from small-variance asymptotics, we consider the MAP estimate obtained by maximizing the joint likelihood with respect to the parameters asymptotically as $\sigma^2$ goes to zero ($\beta$ goes to $\infty$). In our case, it is particularly simple given the joint likelihood above. The log-likelihood easily yields the following asymptotically:

$$\max_{Z,\boldsymbol{\mu},T} - \left( \sum_{t=1}^{N} \| \boldsymbol{x}_t - \boldsymbol{\mu}_{z_t} \|_2^2 + \lambda \sum_{t=2}^{N} \text{KL}(\boldsymbol{z}_t, \boldsymbol{m}_{\boldsymbol{z}_{t-1}}) \right) \tag{4}$$

or equivalently,

$$\min_{Z,\boldsymbol{\mu},T} \left( \sum_{t=1}^{N} \| \boldsymbol{x}_t - \boldsymbol{\mu}_{z_t} \|_2^2 + \lambda \sum_{t=2}^{N} \text{KL}(\boldsymbol{z}_t, \boldsymbol{m}_{\boldsymbol{z}_{t-1}}) \right). \tag{5}$$

Note that, as mentioned above, $\boldsymbol{m}_j = \left\{ T_{jk} \right\}_{k=1}^{K}$. We can view the above objective function as a "penalized" k-means problem, where the penalties are given by the transitions from state to state.

One possible strategy to minimize (5) would be to iteratively minimize with respect to each of the individual parameters $(Z, \boldsymbol{\mu}, T)$ keeping the other two fixed. When fixing $\boldsymbol{\mu}$ and $T$, and taking $\lambda = 1$, the solution for $Z$ in (4) is identical to the MAP update on the latent variables $Z$ for this model, as in a standard HMM. When $\lambda \neq 1$, a simple generalization of the standard forward-backward routine can be used to find the optimal assignments. Keeping $Z$ and $T$ fixed, the update on $\boldsymbol{\mu}_k$ is easily seen to be the equiweighted average of the data points which have been assigned to latent state $k$ in the MAP estimate (it is the same minimization as in k-means for updating cluster means). Finally, since $\text{KL}(\boldsymbol{z}_t, \boldsymbol{m}_j) \propto - \sum_{k=1}^{K} \log T_{jk}$, minimization with respect to $T$ simply yields the empirical transition probabilities, that is $T_{jk,new} = \frac{\text{\# of transitions from state } j \text{ to } k}{\text{\# of transitions from state } j}$, both counts from the MAP path computed during maximization w.r.t $Z$. We observe that, when $\lambda = 1$, the iterative minimization algorithm to solve (5) is exactly the segmental k-means algorithm, also known as Viterbi re-estimation.

### 2.1.3  EM algorithm asymptotics

We can reach the same algorithm alternatively by writing down the steps of the EM algorithm and exploring the small-variance limit of these steps, analogous to the approach of [8] for a Dirichlet process mixture. Given space limitations (and the fact that the resulting algorithm is identical, as expected), a more detailed discussion can be found in the supplementary material.

## 3  Asymptotics of the Infinite Hidden Markov Model

We now tackle the more complex nonparametric model. We will derive the objective function directly as in the parametric case but, unlike the parametric version, we will not apply asymptotics to the existing sampler algorithms. Instead, we will present a new algorithm to optimize our derived objective function. By deriving an algorithm directly, we ensure that our method takes advantage of dynamic programming, unlike the standard sampler.

### 3.1  The Model

The iHMM, also known as the HDP-HMM [3, 12] is a nonparametric Bayesian extension to the HMM, where an HDP prior is used to allow for an unspecified number of states. The HDP is a set of Dirichlet Processes (DPs) with a shared base distribution, that are themselves drawn from a Dirichlet process [12]. Formally, we can write $G_k \sim \text{DP}(\alpha, G_0)$ with a shared base distribution $G_0 \sim \text{DP}(\gamma, H)$, where $H$ is the global base distribution that permits sharing of probability mass across $G_k$s. $\alpha$ and $\gamma$ are the concentration parameters for the $G_k$ and $G_0$ measures, respectively.

To apply HDPs to sequential data, the iHMM can be formulated as follows:

$$\beta \sim \text{GEM}(\gamma), \quad \boldsymbol{\pi}_k | \beta \sim \text{DP}(\alpha, \beta), \quad \theta_k \sim H,$$

$$\boldsymbol{z}_t | \boldsymbol{z}_{t-1} \sim \text{Mult}(\boldsymbol{\pi}_{\boldsymbol{z}_{t-1}}), \qquad \boldsymbol{x}_t \sim \Phi(\theta_{\boldsymbol{z}_t}).$$

For a full Bayesian treatment, Gamma priors are placed on the concentration parameters (though we will not employ such priors in our asymptotic analysis).

Following the discussion in the parametric case, our goal is to write down the full joint likelihood in the above model. As discussed in [12], the Hierarchical Dirichlet Process yields assignments that follow the Chinese Restaurant Franchise (CRF), and thus the iHMM model additionally incorporates a term in the joint likelihood involving the prior probability of a set of state assignments arising from the CRF. Suppose an assignment of observations to states has $K$ different states (i.e., number of restaurants in the franchise), $s_i$ is the number of states that can be reached from state $i$ in one step (i.e., number of tables in each restaurant $i$), and $n_i$ is the number observations in each state $i$ (i.e., number of customers in each restaurant). Then the probability of an assignment in the HDP can be written as (after integrating out mixture weights [1, 11], and if we only consider terms that would not be constants after we do the asymptotic analysis [5]):

$$ p(\mathcal{Z}|\alpha, \gamma, \lambda) \propto \gamma^{K-1} \frac{\Gamma(\gamma + 1)}{\Gamma(\gamma + \sum_{k=1}^{K} s_k)} \prod_{k=1}^{K} \alpha^{s_k - 1} \frac{\Gamma(\alpha + 1)}{\Gamma(\alpha + n_i)}. $$

For the likelihood, we follow the same assumption as in the parametric case: the observation densities are Gaussians with a shared covariance matrix $\sigma^2 I_d$. Further, the means are drawn independently from the prior $\mathcal{N}(0, \rho^2 I_d)$, where $\rho^2 > 0$ (this is needed, as the model is fully Bayesian now). Therefore, $p(\boldsymbol{\mu}_{1:K}) = \prod_{k=1}^{K} \mathcal{N}(\boldsymbol{\mu}_k | 0, \rho^2 I_d)$, and

$$ p(\mathcal{X}, \mathcal{Z}) \propto p(\mathcal{Z}|\alpha, \gamma, \lambda) \cdot p(\boldsymbol{z}_1) \prod_{t=2}^{N} p(\boldsymbol{z}_t | \boldsymbol{z}_{t-1}) \cdot \prod_{t=1}^{N} \mathcal{N}(\boldsymbol{x}_t | \boldsymbol{\mu}_{z_t}, \sigma^2 I_d) \cdot p(\boldsymbol{\mu}_{1:K}). $$

Now, we can perform the small-variance analysis on the generative model. In order to retain the impact of the hyperparameters $\alpha$ and $\gamma$ in the asymptotics, we can choose some constants $\lambda_1, \lambda_2 > 0$ such that
$$ \alpha = \exp(-\lambda_1 \beta), \ \gamma = \exp(-\lambda_2 \beta), $$
where $\beta = 1/(2\sigma^2)$ as before. Note that, in this way, we have $\alpha \to 0$ and $\gamma \to 0$ when $\beta \to \infty$.

We now can consider the objective function for maximizing the generative probability as we let $\beta \to \infty$. This gives $p(\mathcal{X}, \mathcal{Z}) \propto$

$$ \exp\left[ -\beta \left( \sum_{t=1}^{N} \|\boldsymbol{x}_t - \boldsymbol{\mu}_{z_t}\|^2 + \lambda \sum_{t=2}^{N} \mathrm{KL}(\boldsymbol{z}_t, \boldsymbol{m}_{\boldsymbol{z}_{t-1}}) + \lambda_1 \sum_{k=1}^{K} (s_k - 1) + \lambda_2 (K - 1) \right) \right. $$
$$ \left. + \log(p(\boldsymbol{z}_1)) \right]. \tag{6} $$

Therefore, maximizing the generative probability is asymptotically equivalent to the following optimization problem:

$$ \min_{K, Z, \boldsymbol{\mu}, T} \quad \sum_{t=1}^{N} \|\boldsymbol{x}_t - \boldsymbol{\mu}_{z_t}\|^2 + \lambda \sum_{t=2}^{N} \mathrm{KL}(\boldsymbol{z}_t, \boldsymbol{m}_{\boldsymbol{z}_{t-1}}) + \lambda_1 \sum_{k=1}^{K} (s_k - 1) + \lambda_2 (K - 1). \tag{7} $$

In words, this objective seeks to minimize a penalized k-means objective, with three penalties. The first is the same as in the parametric case—a penalty based on the transitions from state to state. The second penalty penalizes the number of transitions out of each state, and the third penalty penalizes the total number of states. Note this is similar to the objective function derived in [8] for the HDP, but here there is no dependence on any particular samplers. This can also be considered as MAP estimation of the parameters, since $p(\mathcal{Z}, \boldsymbol{\mu}|\mathcal{X}) \propto p(\mathcal{X}|\mathcal{Z})p(\mathcal{Z})p(\boldsymbol{\mu})$.

## 3.2 Algorithm

The algorithm presented in [8] could be almost directly applied to (7) but it neglects the sequential characteristics of the model. Instead, we present a new algorithm to directly optimize (7). We follow the alternating minimization framework as in the parametric case, with some slight tweaks. Specifically, given observations $\{\boldsymbol{x}_1, \dots, \boldsymbol{x}_N\}, \lambda, \lambda_1, \lambda_2$, our high-level algorithm proceeds as follows:

(1) Initialization: initialize with one hidden state. The parameters are therefore $K = 1, \boldsymbol{\mu}_1 = \frac{1}{N} \sum_{i=1}^{N} \boldsymbol{x}_i, T = 1$.
(2) Perform a forward-backward step (via approximate dynamic programming) to update $Z$.

(3) Update $K, \boldsymbol{\mu}, T$.

(4) For each state $i$, $(i = 1, \ldots, K)$, check if the set of observations to any state $j$ that are reached by transitioning out of $i$ can form a new dedicated hidden state and lower the objective function in the process. If there are several such moves, choose the one with the maximum improvement in objective function.

(5) Update $K, \boldsymbol{\mu}, T$.

(6) Iterate steps (2)-(5) until convergence.

There are two key changes to the algorithm beyond the standard parametric case. In the forward-backward routine (step 2), we compute the usual $K \times N$ matrix $\alpha$, where $\alpha(c, t)$ represents the minimum cost over paths of length $t$ from the beginning of the sequence and that reach state $c$ at time step $t$. We use the term "cost" to refer to the sum of the distances of points to state means, as well as the additive penalties incurred. However, to see why it is difficult to compute the exact value of $\alpha$ in the nonparametric case, suppose we have computed the minimum cost of paths up to step $t - 1$ and we would like to compute the values of $\alpha$ for step $t$. The cost of a path that ends in state $c$ is obtained by examining, for all states $i$, the cost of a path that ends at $i$ at step $t - 1$ and then transitions to state $c$ at step $t$. Thus, we must consider the transition from $i$ to $c$. If there are existing transitions from state $i$ to state $c$, then we proceed as in a standard forward-backward algorithm. However, we are also interested in two other cases—one where there are no existing transitions from $i$ to $c$ but we consider this transition along with a penalty $\lambda_1$, and another where an entirely new state is formed and we pay a penalty $\lambda_2$. In the first case, the standard forward-backward routine faces an immediate problem, since when we try to compute the cost of the path given by $\alpha(c, t)$, the cost will be infinite as there is a $-\log(0)$ term from the transition probability. We must therefore alter the forward-backward routine, or there will never be new states created nor transitions to an existing state which previously had no transitions. The main idea is to derive and use bounds on how much the transition matrix can change under the above scenarios. As long as we can show that the values we obtain for $\alpha$ are upper bounds, then we can show that the objective function will decrease after the forward-backward routine, as the existing sequence of states is also a valid path (with no new incurred penalties).

The second change (step 4) is that we adopt a "local move" analogous to that described for the hard HDP in [8]. This step determines if the objective will decrease if we create a new global state in a certain fashion; in particular, for each existing state $j$, we compute the change in objective that occurs when data points that transition from $j$ to some state $k$ are given their own new global state. By construction this step decreases the objective.

Due to space constraints, full details of the algorithm, along with a local convergence proof, are provided in the supplementary material (section B).

## 4  Experiments

We conclude with a brief set of experiments designed to highlight the benefits of our approach. Namely, we will show that our methods have benefits over the existing parametric and non-parametric HMM algorithms in terms of speed and accuracy.

**Synthetic Data.** First we compare our nonparametric algorithm with the Beam Sampler for the iHMM[1]. A sequence of length 3000 was generated over a varying number of hidden states with the all-zeros transition matrix except that $T_{i,i+1} = 0.8$ and $T_{i,i+2} = 0.2$ (when $i + 1 > K$, the total number of states, we choose $j = i + 1 \bmod K$ and let $T_{i,j} = 0.8$, and similarly for $i + 2$). Observations were sampled from symmetric Gaussian distributions with means of $\{3, 6, \ldots, 3K\}$ and a variance of 0.9.

The data described above were trained using our nonparametric algorithm (asymp-iHMM) and the Beam sampler. For our nonparametric algorithm, we performed a grid search over all three parameters and selected the parameters using a heuristic (see the supplementary material for a discussion of this heuristic). For the Beam sampling algorithm, we used the following hyperparameter settings: gamma hyperpriors $(4, 1)$ for $\alpha$, $(3, 6)$ for $\gamma$, and a zero mean normal distribution for the base $H$ with the variance equal to 10% of the empirical variance of the dataset. We also normalized the sequence to have zero mean. The number of selected samples was varied among 10, 100, and 1000

for different numbers of states, with 5 iterations between two samples. (Note: there are no burn-in iterations and all samplers are initialized with a randomly initialized 20-state labeling.)

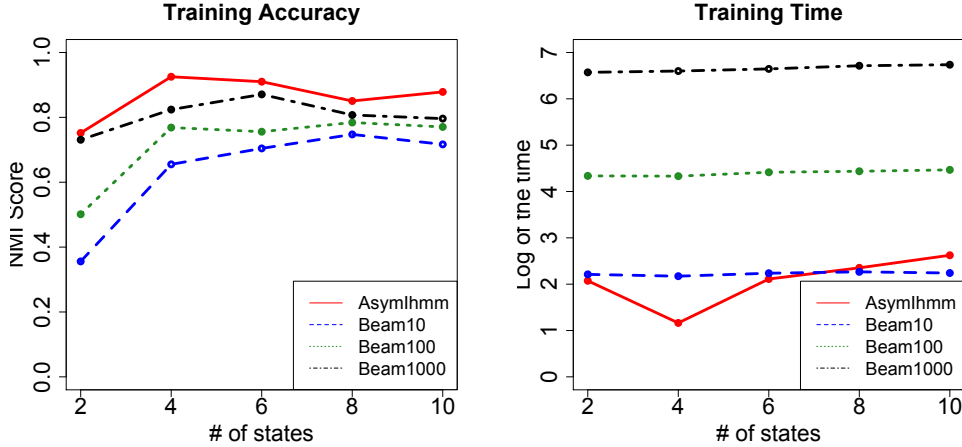

Figure 1: Our algorithm (asymp-iHMM) vs. the Beam Sampler on the synthetic Gaussian hidden Markov model data. (Left) The training accuracy; (Right) The training time on a log-scale.

In Figure 1 (best viewed in color), the training accuracy and running time for the two algorithms are shown, respectively. The accuracy of the Beam sampler is given by the highest among all the samples selected. The accuracy is shown in terms of the *normalized mutual information* (NMI) score (in the range of [0,1]), since the sampler may output different number of states than the ground truth and NMI can handle this situation. We can see that, in all datasets, our algorithm performs better than the sampling method in terms of accuracy, but with running time similar to the sampler with only 10 samples. For these datasets, we also observe that the EM algorithm for the standard HMM (not reported in the figure) can easily output a smaller number of states than the ground truth, which yields a smaller NMI score. We also observed that the Beam sampler is highly sensitive to the initialization of hyperparameters. Putting flat priors over the hyperparameters can ameliorate the situation, but also substantially increases the number of samples required.

Next we demonstrate the effect of the compensation parameter $\lambda$ in the parametric asymptotic model, along with comparisons to the standard HMM. We will call the generalized segmental k-means of Section 2 the 'asymp-HMM' algorithm, shortened to 'AHMM' as appropriate. For this experiment, we used univariate Gaussians with means at 3, 6, and 10, and standard deviation of 2.9. In our ground-truth transition kernel, state $i$ had an 80% prob. of transitioning to state $i+1$, and 10% prob. of transitioning to each of the other states. 5000 datapoints were generated from this model. The first 40% of the data was used for training, and the remaining 60% for prediction. The means in both the standard HMM and the asymp-HMM were initialized by the centroids learned by k-means from the training data. The transition kernels were initialized randomly. Each algorithm was run 50 times; the averaged results are shown in Figure 2.

Figure 2 shows the effect of $\lambda$ on accuracy as measured by NMI and scaled prediction error. We see the expected tradeoff: for small $\lambda$, the problem essentially reduces to standard k-means, whereas for large $\lambda$ the observations are essentially ignored. For $\lambda = 1$, corresponding to standard segmental k-means, we obtain results similar to the standard HMM, which obtains an NMI of .57 and error of 1.16. Thus, the parametric method offers some added flexibility via the new $\lambda$ parameter.

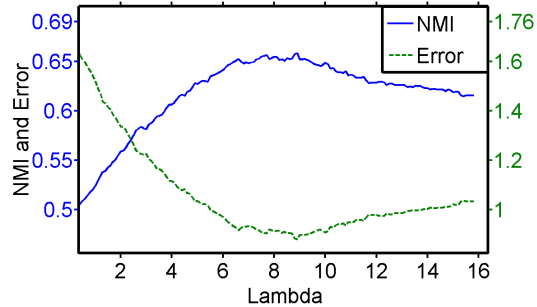

Figure 2: NMI and prediction error as a function of the compensation parameter $\lambda$

**Financial time-series prediction.** Our next experiment illustrates the advantages of our algorithms in a financial prediction problem. The sequence consists of 3668 values of the Standard & Poor's 500 index on consecutive trading days

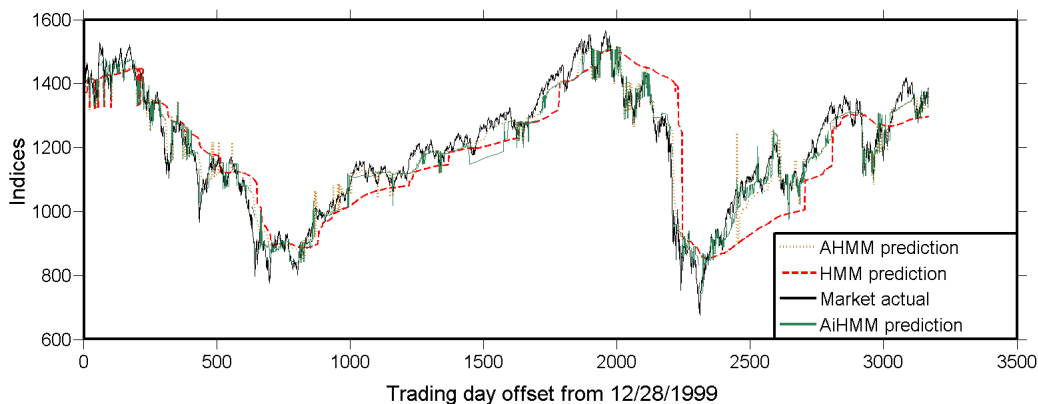

Figure 3: Predicted values of the S&P 500 index from 12/29/1999 to 07/30/2012 returned by the asymp-HMM, asymp-iHMM and the standard HMM algorithms, with the true index values for that period (better in color); see text for details.

from Jan 02, 1998 to July 30, 2012[2]. The index exhibited appreciable variability in this period, with both bull and bear runs. The goal here was to predict the index value on a test sequence of trading days, and compare the accuracies and runtimes of the algorithms.

To prevent overfitting, we used a training window of length 500. This window size was empirically chosen to provide a balance between prediction accuracy and runtime. The algorithms were trained on the sequence from index $i$ to $i + 499$, and then the $i + 500$-th value was predicted and compared with the actual recorded value at that point in the sequence. $i$ ranged from 1 to 3168. As before, the mixture means were initialized with k-means and the transition kernels were given random initial values. For the asymp-HMM and the standard HMM, the number of latent states was empirically chosen to be 5. For the asymp-iHMM, we tune the parameters to get also 5 states on average. For predicting observation $T + 1$ given observations up to step $T$, we used the weighted average of the learned state means, weighted by the transition probabilities given by the state of the observation at time $T$.

We ran the standard HMM along with both the parametric and non-parametric asymptotic algorithms on this data (the Beam sampler was too slow to run over this data, as each individual prediction took on the order of minutes). The values predicted from time step 501 to 3668 are plotted with the true index values in that time range in Figure 3. Both the parametric and non-parametric asymptotic algorithms perform noticably better than the standard HMM; they are able to better approximate the actual curve across all kinds of temporal fluctuations. Indeed, the difference is most stark in the areas of high-frequency oscillations. While the standard HMM returns an averaged-out prediction, our algorithms latch onto the underlying behavior almost immediately and return noticably more accurate predictions. Prediction accuracy has been measured using the mean absolute percentage (MAP) error, which is the mean of the absolute differences of the predicted and true values expressed as percentages of the true values. The MAP error for the HMM was 6.44%, that for the asymp-HMM was 3.16%, and that for the asymp-iHMM was 2.78%. This confirms our visual perception of the asymp-iHMM algorithm returning the best-fitted prediction in Figure 3.

**Additional Real-World Results.** We also compared our methods on a well-log data set that was used for testing the Beam sampler. Due to space constraints, further discussion of these results is included in the supplementary material.

## 5   Conclusion

This paper considered an asymptotic treatment of the HMM and iHMM. The goal was to obtain non-probabilistic formulations inspired by the HMM, in order to expand small-variance asymptotics to sequential models. We view our main contribution as a novel dynamic-programming-based algorithm for sequential data with a non-fixed number of states that is derived from the iHMM model.

## Acknowledgements

This work was supported by NSF award IIS-1217433.

## Footnotes

[1]http://mloss.org/software/view/205/

[2]http://research.stlouisfed.org/fred2/series/SP500/downloaddata

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
