[Supplementary Material · breg_nips2013_supp.pdf]

# Appendices

This document contains supplementary material for the NIPS paper "Small-Variance Asymptotics for Hidden Markov Models."

## A  Parametric HMM Details

In this section we describe the derivation of equation (2) in section 2.1.1 in the main paper, and also discuss the asymptotic treatment of the EM algorithm for the finite-state HMM.

### A.1  Bregman Divergence Transition Representation

First we rewrite the relevant equations from section 2.1.1. We wrote the multinomial distribution for the latent state at time step t as

$$\Pr(\boldsymbol{z}_t | z_{t-1,j} = 1) = \prod_{k=1}^{K} T_{jk}^{z_{tk}}.$$

Using standard algebraic manipulations we can rewrite this as

$$\exp\left[\sum_{k=1}^{K} \log T_{jk}^{z_{tk}}\right] = \exp\left[\sum_{k=1}^{K-1} z_{tk} \cdot \log T_{jk} - z_{tK} \cdot \log \frac{1}{T_{jK}}\right].$$

Recall that, in our binary 1-of-K notation, $z_{tK} = 1 - \sum_{k=1}^{K-1} z_{tk}$. This allows us to rewrite the expression above as

$$\exp\left[\sum_{k=1}^{K} \log T_{jk}^{z_{tk}}\right] = \exp\left[\sum_{k=1}^{K-1} z_{tk} \cdot \frac{\log T_{jk}}{\log T_{jK}} - \log \frac{1}{T_{jK}}\right]. \tag{1}$$

To reparameterize this in exponential family notation, we denote $\boldsymbol{\theta}_j^{-K} = \{\log(T_{jk}/T_{jK})\}_{k=1}^{K-1}$. Then the summation in (1) is clearly the inner product of $\boldsymbol{z}_t^{-K}$ and $\boldsymbol{\theta}_j^{-K}$. This then allows us to re-write (1) as $\exp\left(\langle \boldsymbol{z}_t^{-K}, \boldsymbol{\theta}_j^{-K} \rangle - \psi(\boldsymbol{\theta}_j^{-K})\right)$, where the log-partition function $\psi\left(\boldsymbol{\theta}_j^{-K}\right) = \log(1/T_{jK})$.

Now we will show that the expectation parameter given by $\nabla\psi\left(\boldsymbol{\theta}_j^{-K}\right)$ is exactly the transition probability distribution corresponding to state $j$. From this we will be able to show the Legendre dual of the expectation parameter to be the negative entropy of the transition distribution of $j$. To see this, note that the log-partition function may be written as

$$\psi\left(\boldsymbol{\theta}_j^{-K}\right) = \log \frac{1}{T_{jK}} = \log\left(\frac{T_{jK} + \sum_{k=1}^{K-1} T_{jk}}{T_{jK}}\right) = \log\left(1 + \sum_{k=1}^{K-1} e^{\theta_{jk}^{-K}}\right).$$

We can therefore write the expectation parameter as

$$\boldsymbol{m}_j = \nabla\psi(\boldsymbol{\theta}_j^{-K}) = \nabla \log\left(1 + \sum_{k=1}^{K-1} e^{\theta_{jk}^{-K}}\right) = \left\{\frac{e^{\theta_{jk}^{-K}}}{1 + \sum_{k=1}^{K-1} e^{\theta_{jk}^{-K}}}\right\}_{k=1}^{K-1} = \left\{T_{jk}\right\}_{k=1}^{K-1}.$$

Then the Legendre dual $\phi$ can be written as

$$\phi(\boldsymbol{m}_j) = \langle \boldsymbol{m}_j, \boldsymbol{\theta}_j^{-K} \rangle - \psi(\boldsymbol{\theta}_j^{-K}) = \sum_{k=1}^{K-1} T_{jk} \log \frac{T_{jk}}{T_{jK}} + \log T_{jK} = \sum_{k=1}^{K} T_{jk} \log T_{jk}.$$

Note that this is exactly the negative entropy of the transition probability distribution $\boldsymbol{\pi}_j = T_{j\cdot} = \left\{T_{jk}\right\}_{k=1}^{K}$.

The Bregman divergence derived from $\phi$ is therefore

$$
\begin{aligned}
d_\phi(\boldsymbol{z}_t, \boldsymbol{m}_j) &= \phi(\boldsymbol{z}_t) - \phi(\boldsymbol{m}_j) - \langle \boldsymbol{z}_t - \boldsymbol{m}_j, \nabla\phi(\boldsymbol{m}_j)\rangle \\
&= \sum_{k=1}^{K} z_{tk}\log z_{tk} - \sum_{k=1}^{K} T_{jk}\log T_{jk} - \sum_{k=1}^{K}(z_{tk} - T_{jk})\left(1 + \log T_{jk}\right) \\
&= \sum_{k=1}^{K} z_{tk}\log\frac{z_{tk}}{T_{jk}} = \mathrm{KL}(\boldsymbol{z}_t, \boldsymbol{m}_j),
\end{aligned}
$$

since we have $\sum_{k=1}^{K} z_{tk} = 1$ and $\sum_{k=1}^{K} T_{jk} = 1$.

Then, by the bijection established in Banerjee et al. [1], we can write the distribution $\Pr(\boldsymbol{z}_t|z_{t-1,j} = 1)$ as

$$
\Pr(\boldsymbol{z}_t|z_{t-1,j} = 1) = \exp\left(\langle \boldsymbol{z}_t^{-K}, \boldsymbol{\theta}_j^{-K}\rangle - \psi(\boldsymbol{\theta}_j^{-K})\right) = \exp(-d_\phi(\boldsymbol{z}_t, \boldsymbol{m}_j))b_\phi(\boldsymbol{z}_t),
$$

where the $\boldsymbol{m}$-independent function $b_\phi(\boldsymbol{z}_t)$ is

$$
b_\phi(\boldsymbol{z}_t) = \exp(\phi(\boldsymbol{z}_t)) = \exp\left(\sum_{k=1}^{K} z_{tk}\log z_{tk}\right) = 1,
$$

where we have again used the binary 1-of-K representation of $\boldsymbol{z}_t$. This completes the derivation of equation (2) in section 2.1.1 of the paper.

Once we are given this appropriate Bregman divergence representation, we can directly apply the recipe in [3], Lemma 3.1. In particular, the variance on an exponential family distribution may be scaled by appropriately scaling the partition function and the underlying natural parameter. Stated simply, we can introduce a new parameter $\hat\beta$, as in Jiang et al., and use a scaled version of the Bregman divergence representation, i.e., $\exp(-\hat\beta d_\phi(\boldsymbol{z}_t, \boldsymbol{m}_j))b_{\hat\beta\phi}(\boldsymbol{z}_t)$.

## A.2 EM Algorithm Asymptotics

Here we discuss the asymtotic behavior of the EM algorithm for the standard HMM, and show its reduction to segmental K-means.

The reduction of the E-step posterior marginal update to the MAP estimate is fairly straightforward but involved. Let us define $\gamma(z_{tk})$ as the probability of the value of the $t^{th}$ hidden variable being $k$, given the observations, the means, and the $T$ matrix. This probability can be expressed as

$$
\gamma(z_{tk}) = \frac{\alpha(z_{tk})\beta(z_{tk})}{\sum_{l=1}^{K}\alpha(z_{tl})\beta(z_{tl})} = \frac{1}{1 + \sum_{l=1;l\neq k}^{K}\frac{\alpha(z_{tl})\beta(z_{tl})}{\alpha(z_{tk}),\beta(z_{tk})}} \tag{2}
$$

where $\alpha(\boldsymbol{z}_t) = p(\boldsymbol{x}_1, ..., \boldsymbol{x}_t, \boldsymbol{z}_t)$ and $\beta(\boldsymbol{z}_t) = p(\boldsymbol{x}_{t+1}, ..., \boldsymbol{x}_N \,|\, \boldsymbol{z}_t)$; the $\gamma$ probabilities are what are computed by the E-step of the EM algorithm for the standard HMM. Asymptotically as $\sigma^2 \to 0$, we find that this probability goes to zero for all states $k$ except the one that falls on the MAP path, which may be derived by expanding the ratios of $\alpha$ and $\beta$ probabilities in the above denominator. Thus in the E-step analogue of our algorithm, we first find the MAP path, and for each latent state on that path, we set the corresponding (temporal) posterior probability to 1 and that for the other states to 0. Note that the MAP path for the $Z$ variables is obtained using a standard Viterbi dynamic programming algorithm with a forward and backward pass.

To complete the derivation of segmental k-means, we now consider the updates to $T$ and $\boldsymbol{\mu}$ in the M-step analog. Recall the expected log-likelihood equation used in the M-step (see, e.g., Bishop [2]):

$$\mathcal{Q}(\Theta, \Theta_{old}) = \sum_{k=1}^{K} \gamma(z_{1k}) \ln \pi_k + \sum_{n=1}^{N} \sum_{k=1}^{K} \gamma(z_{nk}) \ln p(\mathbf{x}_n | \boldsymbol{\mu}_k)$$
$$+ \sum_{n=2}^{N} \sum_{j=1}^{K} \sum_{k=1}^{K} \xi(z_{n-1,j}, z_{nk}) \log T_{jk}.$$

Here $\xi(z_{n-1,j}, z_{nk})$ is the joint posterior marginal of pairwise latent variables, and $\Theta$ and $\Theta_{old}$ represent the new and existing values, respectively, of $Z$, $\boldsymbol{\mu}$ and $T$. We first consider maximizing $\mathcal{Q}(\Theta, \Theta_{old})$ with respect to the transition probabilities.

It can be noted that since we have hard cluster assignments in the asymptotic E-step, the joint posterior marginals will also be binary, and therefore $\sum_{n=1}^{N} \xi(z_{n-1,j}, z_{nk})$ represents the total number of times the transitions from state $j$ to state $k$ has occurred in the entire chain. Thus, for each state $j$, the maximization of the expected log-likelihood is a linear program over the $\log T_{jk}$s, and the solution can be seen to be clearly the empirical transition probabilities computed from the MAP transition sequence.

The final step is the updates to the means. This straightforwardly can be derived to be an update of the empirical means of all points assigned to each state in a manner similar to the derivation of the k-means cluster mean update as a limit of the M-step in a Gaussian mixture model. This completes the derivation of segmental k-means as a small-variance approximation of the EM algorithm in the presence of exponential family probabilities.

## B  Algorithm for the infinite Hidden Markov Model

Here, we give the detailed description of the algorithm in Section 3.2. The optimization problem we aim to solve is

$$\min_{K,z,\boldsymbol{\mu},T} \quad \sum_{t=1}^{N} \|\boldsymbol{x}_t - \boldsymbol{\mu}_{z_t}\|^2 - \lambda \sum_{t=2}^{N} \log T_{z_{t-1},z_t} + \lambda_1 \sum_{k=1}^{K} (s_k - 1) + \lambda_2 (K - 1), \qquad (3)$$

where $\{\boldsymbol{x}_1, \ldots, \boldsymbol{x}_N\}$ are the observations, $\{z_1, \ldots, z_N\}$ is the hidden state sequence, $K$ is the total number of hidden states, $T$ is the transition probability matrix with $T_{i,j} = \Pr(z_{t+1} = j | z_t = i)$, and $s_i (i = 1, \ldots, K)$ is the number of states that can be reachable from state $i$. Note that we are expressing the transition penalties here in terms of $T$ as opposed to the KL divergence, for ease of presentation.

To optimize (3), we follow an alternating minimization framework. We first determine the sequence of states to optimize the objective when all but $Z$ is fixed using a forward-backward routine. Here, we cannot apply exact dynamic programming, due to the possible creation of new states as well as the change in transition probabilities when either creating a new state or paying a $\lambda_1$ penalty. We then update the means of each state as the empirical means based on the state assignments, and the transition matrix as the empirical transition matrix. We further adopt a move analogous to that described for the hard HDP in [4]. This step determines if the objective will decrease if we create a new hidden state in a certain fashion; in particular, for each existing state $j$, we compute the change in objective that occurs when observations that transition from state $j$ to some state $k$ are given their own new hidden state.

Specifically, given observations $\{\boldsymbol{x}_1, \ldots, \boldsymbol{x}_N\}, \lambda, \lambda_1, \lambda_2$, our high-level algorithm proceeds as follows:

(1) Initialization: initialize with one hidden state. The parameters are therefore $K = 1, \boldsymbol{\mu}_1 = \frac{1}{N} \sum_{i=1}^{N} \boldsymbol{x}_i, T = 1$.

(2) Perform a forward-backward step (via approximate dynamic programming) to update $Z$.

(3) Update $K, \boldsymbol{\mu}, T$.

(4) For each state $i$, $(i = 1, \ldots, K)$, check if the set of observations to any state $j$ that are reached by transitioning out of $i$ can form a new dedicated hidden state and lower the objective function in the process. If there are several such moves, choose the one with the maximum improvement in objective function.

(5) Update $K, \boldsymbol{\mu}, T$.

(6) Iterate steps (2)-(5) until convergence.

Note: Step (3) here is only for clearer presentation of step (4).

## B.1 Forward-Backward Step

In the forward-backward step, we will compute a $K \times N$ matrix $\alpha$, where $\alpha(c, t)$ represents the minimum cost over paths of length $t$ from the beginning of the sequence and that reach state $c$ at time step $t$. We use the term "cost" to refer to the sum of the distances of points to state means, as well as the additive penalties incurred. Since we are interested in the possibility of potentially creating new states during this forward-backward process, and the creation of new states will necessarily change the transition probabilities, it does not appear that $\alpha$ can be computed exactly. We instead describe a procedure that computes an upper bound for each value of $\alpha$.

To give further intuition for why it is difficult to compute the exact value of $\alpha$: suppose we have computed the minimum cost of paths up to step $t - 1$ and we would like to compute the values of $\alpha$ for step $t$. The value of a path that ends in state $c$ is obtained by examining, for all states $i$, the cost of a path that ends at $i$ at step $t - 1$ and then transitions to state $c$ at step $t$. Thus, we must consider the transition from $i$ to $c$. If there are existing transitions from state $i$ to state $c$, then we proceed as in a standard forward-backward algorithm. However, we are also interested in two other cases—one where there are no existing transitions from $i$ to $c$ but we consider this transition along with a penalty $\lambda_1$, and another where an entirely new state is formed and we pay a penalty $\lambda_2$. In the first case, the standard forward-backward routine faces an immediate problem, since when we try to compute the cost of the path given by $\alpha(c, t)$, the cost will be infinite as there is a $-\log(0)$ term from the transition probability. We must therefore alter the forward-backward routine, or there will never be new states created nor transitions to an existing state which previously had no transitions. The main idea is to derive and use bounds on how much the transition matrix can change under the above scenarios. As long as we can show that the values we obtain for $\alpha$ are upper bounds, then we can show that the objective function will decrease after the forward-backward routine, as the existing sequence of states is also a valid path (with no new incurred penalties). That is, the cost we compute here is an upper bound of both the non-penalized objective function value and newly-introduced penalties. If there is no new states created or no new transitions happened, the upper bound for the newly-introduced penalties is zero as mentioned before.

Now we describe the algorithm in more detail. In the following description, $old\_K$ is the number of states from the previous iteration, and $K$ is the number of states before the computation of $\alpha$ at time step $t$. Then, for each state $1 \le c \le K$,

(I). If we are transitioning from a state $i \le old\_K$,

- Standard situation: If $c \le old\_K$ and $T_{i,c} \neq 0$, we compute (as in the parametric case):

$$d(i, c) = \|\boldsymbol{x}_t - \boldsymbol{\mu}_c\|^2 - \lambda \times \log(T_{i,c}). \tag{4}$$

- No existing transitions: Otherwise, we use the following upper bound when transitioning to state $c$:

$$d(i, c) = \|\boldsymbol{x}_t - \boldsymbol{\mu}_c\|^2 - \lambda \times \log\left(\frac{1}{n_i}\right) + \lambda \times E_i + \lambda_1, \tag{5}$$

where,

- $n_i$ is the total number of transitions out of state $i$.
- $E_i$ is the upper bound of the possible change in one transition probability from state $i$ incurred by adding one state to the reachable pool of state $i$. We compute the change of the maximum of $\frac{n_{ij}}{n_i}$, and we have (assume $j$ is the largest one)

$$E_i = (n_{ij} - 1) \times \left( \log \frac{n_{ij}}{n_i} - \log \frac{n_{ij} - 1}{n_i} \right). \tag{6}$$

    &ndash; $\lambda_1$ is the penalty incurred from transiting to a new state.

(II). Entirely new state: Otherwise, we use the bound:

$$d(i,c) = \|\boldsymbol{x}_t - \boldsymbol{\mu}_c\|^2 - \lambda \times \log\left(\frac{1}{N-1}\right) + \lambda_1. \tag{7}$$

where,

- $\frac{1}{N-1}$ is an upper bound of the transition probability from state $i$ to state $c$,
- $0$ is the upper bound of possible change in the transition probabilities from state $k$ where the original transition occurred for this transition. Here, in this path which involves transition from an entirely new state $i$ to state $c$, we not only add a new row to the transition matrix $T$, but also change another existing row of $T$. That is, the transit-out probability of state $k$ will be changed since it loses one count of transition to state $c$.
- $\lambda_1$ is the penalty incurred when adding this new state transition out of state $i$.

Empirically, we will use $d(i,c) = \|x_t - \mu_c\|^2 + \lambda_1$ since it still decreases the objective function value monotonically most of the time in practice and yields better results.

(III). We compute the minimum among all states:

$$\alpha(c,t) = \min_{1 \leq i \leq K} \alpha(i, t-1) + d(i,c) \tag{8}$$

To check if this time step can be created as a new hidden state, we find

$$d_{\min} = \min_{i,c} d(i,c). \tag{9}$$

If $d_{\min} > \lambda_1 + \lambda_2$, we create a new hidden state (this is the penalty incurred for transiting to a new state and a new hidden state). We let $K = K + 1$,

$$\alpha(K,t) = \min_{1 \leq k \leq K-1} \alpha(k, t-1) + \lambda_1 + \lambda_2. \tag{10}$$

This follows from the description of (II).

Now, we prove the correctness of the upper bounds. For the first bound, when we consider transitioning from $i$ to $c$ such that $T_{ic} = 0$, we are adding a new transition out of state $i$, which then impacts the other transition probabilities out of state $i$. Thus, the $E_i$ bound determines how much the change in the other transition probabilities impacts the current path cost.

**Lemma 1.** *$E_i$ is an upper bound of the possible change in the objective function value, in terms of other transition probabilities from state $i$, incurred by adding one transition to a new state in Step (I). Here, we assume the total number of transitions from state $i$ is fixed.*

*Proof.* Denote $n_i$ as the total number of transitions from state $i$, $n_{ij}$ as the total number of transitions from state $i$ to state $j$ with $n_{ij} > 0$. Thus, the possible change for other transitions is

$$(n_{ij} - 1) \times \left( \log \frac{n_{ij}}{n_i} - \log \frac{n_{ij} - 1}{n_i} \right) \geq 0.$$

Let $f(x) = (x-1)(\log x - \log(x-1)), x \geq 2$, we have

$$f'(x) = \log x - \log(x-1) - \frac{1}{x}$$

$$= \log x - [\log x + \frac{-1}{x} - \frac{1}{2x^2} + o(\frac{1}{2x^2})] - \frac{1}{x}$$

$$= \frac{1}{2x^2} + o(\frac{1}{2x^2}) > 0$$

Thus, $f(x)$ is increasing as $x$ increases for $x \geq 2$. When $x = 1$, we have by definition $f(1) = 0$. Therefore,

$$E_i = (n_{ik} - 1) \times \left( \log \frac{n_{ik}}{n_i} - \log \frac{n_{ik} - 1}{n_i} \right) = \max_j (n_{ij} - 1) \times \left( \log \frac{n_{ij}}{n_i} - \log \frac{n_{ij} - 1}{n_i} \right),$$

where $k = \mathrm{argmax}_j n_{ij}$. $\qquad\square$

The second bound deals with transitioning from an entirely new state. We are adding one row to the transition matrix $T$, which then also changes another row of transition probabilities out of state $k$ where this transition previously is coming from state $k$.

**Lemma 2.** *0 is an upper bound of possible change in the objective function value in terms of the transition probabilities from state $k$ where the original transition occurred for this transition in Step (II).*

*Proof.* Denote $n_k$ the total number of transitions from state $k$, $n_{kj}$ the total number of transitions from state $k$ to state $j$ with $n_{kj} > 0$. Without loss of generality, we assume that the lost transitions are all from $n_{ki}$ and the number of lost transitions is $x$. Thus, the change is

$$\sum_j n_{kj} \log \frac{n_{kj}}{n_k} - \sum_{j \neq i} n_{kj} \log \frac{n_{kj}}{n_k - x} - (n_{ki} - x) \log \frac{n_{ki} - x}{n_k - x}$$

$$= \sum_{j \neq i} n_{kj} \log \frac{n_k - x}{n_k} + n_{ki} \left( \log \frac{n_{ki}}{n_k} - \log \frac{n_{ki} - x}{n_k - x} \right) + x \log \frac{n_{ki} - x}{n_k - x} \leq 0,$$

since $\frac{n_{ki}}{n_k} \leq 1$, $\frac{n_k - x}{n_k} < 1$, $\frac{n_{ki} - x}{n_k - x} \leq 1$, and $\frac{n_{ki}}{n_k} \geq \frac{n_{ki} - x}{n_k - x}$. $\qquad\square$

From Lemma 1 and 2, we know that in each time step we compute an upper bound of the minimum non-penalized objective function value. And since we add $\lambda_1$ and $\lambda_2$ in $\alpha$ whenever we transition to a state where there was no existing transitions or we create an entirely new state respectively, we also manage to upper-bound the newly-introduced penalties. Combining these two together, we have the following:

**Proposition 1.** *The computation of $\alpha$ gives an upper bound of the minimum cost of every possible path.*

## B.2 Local Move - Step 4

After finishing the forward-backward step and updating the means and transition matrix, we check locally if we should create a new state by determining, for all $i$ and $k$, if the objective function is lowered when all data points in state $k$ that reached state $k$ via state $i$ are put into a new state. In particular, we determine and execute the single best such move.

In detail, for each state $1 \leq i \leq K$, we consider all the time steps $A_i$ that are one step from a time step with state $i$. These time steps $A_i$ can be grouped by their states: $A_i(1), \ldots, A_i(K)$, where $A_i(j)$ indicates time steps belonging to state $j$. Then, for all states $j$ with $|A_i(j)| > 0$, we can compute the objective function contribution from these time steps. We have

$$old = \sum_{t \in A_i(j)} \{\|\boldsymbol{x}_t - \boldsymbol{\mu}_{z_t}\|^2 - \lambda \times \log T_{i,z_t} - \lambda \times \log T_{z_t,z_{t+1}}\}. \tag{11}$$

If we let $A_i(j)$ be a new hidden state, the contribution would be

$$new = \sum_{t \in A_i(j)} \{\|\boldsymbol{x}_t - \bar{\boldsymbol{x}}_t\|^2 - \lambda \times \log T_{i,z_t} - \lambda \times \log T_{K+1,z_{t+1}}\} + \lambda_1 \times K_i(j) + \lambda_2, \tag{12}$$

where

- $\bar{\boldsymbol{x}}_t = \frac{1}{|A_i(j)|} \sum_{t \in A_i(j)} \boldsymbol{x}_t$;

- Let $B_i(j)$ be the time steps which are one-step from those of $A_i(j)$. $T_{K+1,z_{t+1}}$ is just the empirical transition probability from time step $t \in A_i(j)$ to the next time step $t+1 \in B_i(j)$.

- $K_i(j)$ is the number of states in $B_i(j)$.

- Since there is no change of $\lambda_1$ penalty for state $i$, we do not need to consider this in the "old" contribution.

If $new < old$, then $A_i(j)$ is considered as a candidate for a new hidden state.

After a whole pass of the states, we find the largest reduction

$$[i, j] \in \text{argmax}_{i,j} \ old_i(j) - new_i(j), \tag{13}$$

and create $A_i(j)$ as a new hidden state.

In summary, in the forward-backward step, the cost we compute is an upper bound of both the non-penalized objective function value and newly-introduced penalties. If there is no new states created or no new transitions happened, the upper bound for the newly-introduced penalties is zero. That is, we preserve the non-penalized objective function value of any path from previous iteration, and upper-bound the non-penalized objective function value and the newly-introduced penalties of any new path we find. In the local move step, we further reduce the objective function value by considering locally for each state we get in the forward-backward step. Therefore, we have

**Proposition 2.** *The algorithm decreases the objective function value in each iteration.*

*Proof.* We know that the best path $old\_P$ obtained from last iteration would be a possible path in this iteration, and from the forward-backward step, it would preserve its cost from the previous iteration. Thus, we have

$$\alpha(new\_P) = \min_{\text{all possible P}} \alpha(P) \leq \alpha(old\_P).$$

From Proposition 1, we know $\alpha$ gives the upper bound of the additional incurred cost, thus

$$\text{cost}(new\_P) \leq \text{cost}(old\_P).$$

We also further decreases the cost from the local move, which preserves the inequality. That is, we have

$$\text{cost}(new\_P) \leq \text{cost}(old\_P).$$

$\square$

## C  Additional Experimental Details

### C.1  Heuristic for Parameter Selection

In this section, we give a simple heuristic which we found empirically promising for parameter selection. We observe empirically that there is a gap between 1 and the smallest resulting number of states larger than 1 (as in Figure 1) when performing a grid search over all the $\lambda$ values. Thus, we select the parameters that yield the smallest number of states bigger than 1 and consider this as the estimate of the number of hidden states. If there are many combinations of parameters that reach the estimated number of states, we choose the set of parameters which gives the best fit, that is, the smallest non-penalized objective function value. We stress that this technique is a heuristic, and do not claim any theoretical justification for it.

### C.2  Well-log data

This additional experiment illustrates qualitative performance of our algorithms on a changepoint detection problem. The data consists of 4050 noisy NMR measurements of rock strata obtained via lowering a probe through a bore-hole. The data has been previously analyzed in [5] by eliminating the forty greatest outliers and running a changepoint detection algorithm with a fixed number of changepoints. This approach works well as this one-dimensional dataset can be inspected visually to make a decision on whether to throw away datapoints and get a rough idea for the number of changepoints. It has been noted in [5] that Gaussian mixtures seem to be a good model for this data; in addition visual inspection of the data seems to suggest five clusters, at five distinct depths. Thus we used a five-state Gaussian HMM to model this data. As in the synthetic case, we first ran k-means for initialization of the mixture means and randomly initialized the transition/pseudo-transition kernels.

The state sequence inferred by our algorithm is color-plotted in Figure 2. The changepoints seem to have been accurately detected, with imperceptibly small noise. Relative performance trends are

Figure 1: An example showing the trend of the number of states as a function of lambda.

Figure 2: Visual display of the state assignments provided by the asymp-HMM algorithm on the well-log data. Each color shows a different state (best viewed in color); see text for details.

similar to the synthetic case; the standard HMM took an average of 16 iterations to converge, with an average running time of 4.5s. Our algorithm converges in 9 iterations, with an average time of 1.7s with $\lambda = 3$.