[Reviews · NeurIPS 2013]

Submitted by Assigned_Reviewer_4

The paper provides a small-variance asymptotics analysis for the standard HMM as well as the infinite HMM, and in particular uses the asymptotic analysis to derive "hard clustering" optimization programs for learning and inference in both kinds of HMMs.

The exposition in the paper is excellent, and it deftly explains the framework and algorithm derivation (with two approaches, providing extra intuition pumps). The paper also explains connections to existing algorithms (for standard HMMs, Viterbi re-estimation corresponds to setting the algorithm parameter lambda to 1) and previous work in small-variance asymptotics. It clearly delineates the challenges in adapting the small-variance algorithm approaches to the (infinite) HMM (Section 3.2) and explains good methods to overcome those challenges. These HMM fitting methods are likely to be very interesting to the community, and the development here (and in the supplementary materials) is really great.


However, the experiments are weak. The data, both real and synthetic, are very easy, and it seems unlikely that the only comparison to existing techniques (the beam sampler) does justice to existing techniques.

Despite the fact that synthetic data is especially easy, the beam sampler does very poorly, yet I ran the same experiment using a library that came up as the first result for 'python hmm sampling' (pyhsmm), which primarily uses weak limit samplers (which have been used in many application papers and certainly seem to fit these experiments well), and got much better results in a fraction of the time. Direct comparison to the results in the paper is hard because there is no formula or citation provided for the NMI performance metric, and the timing plot in Figure 1 has no units (just "log of the time" on the vertical axis), but it seems the weak limit sampler is decoding the synthetic sequences nearly perfectly in just 0.03 seconds per Gibbs sweep on my laptop. Given the papers showing successful applications of weak limit samplers (many times without dependence on good initializations or slow convergence, as mentioned in this paper's abstract and introduction) and the easy availability of code (there are several libraries online), it seems misleading not to provide any comparison or comment, yet this paper only provides a surprisingly weak showing of the beam sampler.

There are other weaknesses and points to clarify in the experiments section:

(1) "For these datasets, we also observe that the EM algorithm for the standard HMM (not reported in the figure) can easily output a smaller number of states than the ground truth, which yields a smaller NMI score." Why not show the results? The synthetic dataset should be easy for EM, so the omission is strange and unexplained.

(2) "Next we demonstrate... along with comparison to the standard HMM." I don't see any lines in Figure 2 corresponding to the standard HMM methods. Were they left out accidentally?

(3) Was the time for the grid search over algorithm parameters (mentioned in the third paragraph of Section 4) included in the reported run times (which seems appropriate, since the iHMM methods are supposedly fitting concentration parameters)? What are the units for the vertical axis in the right-hand panel of Figure 2?

Timing experiments can be a rabbit hole and a line must be drawn somewhere, but it's not clear what can be learned from the given experimental results, and as presented they do not support the paper's relative algorithm performance claims. These issues with the experimental evaluation provided could be easily remedied by toning down the claims about existing methods and focusing instead on the fact that the experiments demonstrate the proposed method is competitive with an iHMM sampling method (and perhaps also with EM if those results were reported).


Despite those issues, the paper remains a great treatment of an exciting new analytical and algorithmic perspective on the well-loved (i)HMM.
Summary: The paper provides a great treatment of a subject of great interest to the NIPS community, though the experiments should be improved or the claims about the experiments adjusted.

Submitted by Assigned_Reviewer_5

The authors use, by now well-established, tools for small-variance
asymptotics of latent-variable model and apply to Hidden Markov Models
(HMM) and their infinite, non-parametric counter parts. For the former
model class, they recover as a special case segmental k-means or
Viterbi re-estimation. For infinite HMM they obtain an alternative
training algorithm which outperforms a fast sampling scheme in their
evaluation.

The methodological results are not very surprising, but constitute a
natural extension of relevance of small-variance asymptotics, given
the wide-spread use of HMM.

The paper has two major weaknesses.

Organization: Given that small-variance asymptotics have been
considered for several types of latent-variables, a more detailed
description of the contribution---mostly 3.2 and also the details in
3.1---would have been been preferable. Also, the quality of the
writing varies. In particular 3.2 is much weaker (and hand-wavy) than
the other parts and gives insufficient to the point of not being
reproducible. A more careful balance of basics, introduction and core
contribution would improve the paper very much.

Evaluation: I disagree with the choice of computational examples
and find the level of details given insufficient. I would have
expected to see an evaluation using real problems in which HMM perform
well and where variable number of states would be needed. I find that
neither computational example is particularly enlightening. For
example, in most applications of HMMs, self-transition probabilities
are not zero and in particular getting the state-durations right is
what helps with a lot of applications with very noisy data in biology
(e.g., ArrayCGH). I don't believe that the examples were cherry-picked
to show the advantage of the proposed method, but they certainly do
not do the authors a favor by persuading scientists using HMM to try
their method.

Also, I would have preferred to see the number of states inferred for
both types of algorithms, and in general more details and further
experiments (for example: what happens for larger k in the experiment
in Figure 1. The beam sampler's running time barely change, while the
running time of the proposed method increases drastically.

A more general comment: It is well known, in the HMM literature at
least, that modeling emissions is very important. There are very fast
samplers for HMM with emissions modeled by infinite mixtures available
(Yau et al. J R Stat Soc Series B Stat Methodol. 2011), which perform
very well at least on the aforementioned ArrayCGH example. It would be
interesting to see, how the proposed method, which I realize could be extended
to more complicated emission distributions, would perform in comparison.
Summary: Natural extension of small-variance asymptotics of latent-variable models to HMM with an evaluation which is unlikely to convince researchers using HMM in real applications.

Submitted by Assigned_Reviewer_8

Recent work by Kulis et al has revisited some latent variable
models to examine their ``small variance asymptotics'' ---
the limiting behavior of their inference algorithms as
variance tends to zero. For example, Gaussian mixture models
have been shown to behave like the k-means algorithm in this
small-variance limit. This paper does the same analysis
for the HMM as well as its Bayesian nonparametric version,
the infinite HMM (with hierarchical Dirichlet prior). The development
follows that of previous work by Jiang et al. and the authors
present intuitive combinatorial algorithms (that look like
penalized versions of k-means) in both the traditional and
infinite settings.

I am not very familiar with the prior work in this thread,
but the result of the current submission seems novel enough
for a NIPS publication. The technical details required
to develop the analysis in the infinite HMM case in particular
are quite nontrivial to work out. Additionally, the writing
was mostly understandable, though I wish that the pseudocode
for Section 3.2 would have been spelled out in some more detail.
In sum, it's a nice contribution and I recommend acceptance.

The weakest part of the paper is the experiments section. The
simulated data experiments are quite small scale and do little
to illustrate the strengths or weaknesses of the approach.
To really show off the efficiency of these combinatorial algorithms,
it seems necessary to tackle a problem with a much larger state
space.

The financial dataset likewise does not seem like a serious
experiment as the number of states used is again 5. We also
don't *really* believe that 5-state HMMs are the `correct' way to
model this dataset right? But my other question about this dataset
is --- why would we expect the combinatorial algorithm to perform
better than the sampling based algorithm? Is there a good rule of
thumb to deciding when one is better than the other? I suppose this
question is a bit like answering when k-means is better than gaussian
mixture models --- and I would think that in many cases, the opposite
is true.

Finally, in light of all the new activity in the community on
spectral methods for latent variable models (see, e.g., Hsu et al. COLT '09),
how do these small-variance algorithms compare? We know that they
can still get stuck in local optima...


Summary: This is a good extension of a line of research on ``small variance asymptotics''. Despite limited experiments, I recommend acceptance.
Author Feedback

Author rebuttal: 